# Development of modified CTAB and Trizol protocols to isolate high molecular weight (HMW) RNA from polyphenol and polysaccharides rich pigeonpea (*Cajanuscajan* (L.) Millsp

**Pawan Mainkar**[1], **Deepanshu Jayaswal**[1,2]*, **Deepesh Kumar**[1], **Kuldip Jayaswall**[2], **Sandeep Jaiswal**[3], **Arvind Nath Singh**[2], **Sanjay Kumar**[2], **Rekha Kansal**[1]*

**1** ICAR-National Institute for Plant Biotechnology, New Delhi, India, **2** ICAR-Indian Institute of Seed Science, Mau, Uttar Pradesh, India, **3** ICAR-Research Complex for NEH region, Umiam, Meghalaya, India

* jayaswaldeepanshu@gmail.com (DJ); rekhakansal@hotmail.com (RK)

## Abstract

Pigeonpea (*Cajanuscajan* L.) is a legume crop that contains high levels of polyphenolic compounds and polysaccharides that become a hindrance in extracting good-quality and enough amount of RNA from its tissues. With the existing methods of RNA isolation, the phenolic compounds may co-precipitate or bind to the RNA giving false results. Therefore, in the present study, we have modified conventional CTAB and Trizol-based methods which resulted in good quality with the absorbance A260/A280 ratios in the range of 1.83 to 1.98 and A260/230 ratios in the range of 2.0–2.23, revealed RNA to be of high purity and free of contaminants. Both of the proposed protocols yielded a good quantity of RNA ranging from 289 to 422μg per gram of tissue. Distinctly visible bands of 28S and 18S rRNA were observed without degradation or smear, which indicated the presence of intact RNA. RT-PCR analysis showed that isolated RNA was quantitatively sufficient and compliant for the subsequent gene expression analysis.

## Introduction

Pigeonpea (*Cajanuscajan* L), is a diploid ($2n = 2x = 22$), legume food crop with an estimated genome size of 858Mb pand 11 pairs of chromosomes [1]. It is widely cultivated by small-holder farmers in all tropical and semitropical regions of India. It is a rich source of dietary proteins, vitamins, and minerals, which playa significant role in nutritional security. Additionally, pigeonpea possesses a significant number of polysaccharides and polyphenols [2, 3]. The phenolic compounds irreversibly interact with nucleic acids (DNA/RNA) and proteins which oxidized to quinones [4, 5]. Polysaccharides co-precipitate with the nucleic acids mainly RNA giving a viscous fluid instead of a solid pellet, impeding the isolation of high-quality RNA which results in decreased RNA yield [6]. Hence, it is crucial to isolate good quality and

**Data Availability Statement:** All relevant data are within the paper and its Supporting Information files.

**Funding:** The author(s) received no specific funding for this work.

**Competing interests:** The authors have declared that no competing interests exist.

quantity of total RNA, which improves the functional genomics, transcriptome analysis, cDNA library preparations, northern blotting, qRT-PCR, and microarray hybridization in pigeonpea for discovering the novel candidate genes and their functions in different kinds of biotic and abiotic stresses.

To date there are several standard protocols reported for isolation of RNA from legumes, viz. Urea-Lithium chloride (LiCl) lysis buffer protocol [7] for Date palm, *Cajanuscajan*, *Dolichosbiflorus* and *Vigna mungo*, sucrose- sodium chloride and LiCl extraction buffer protocol [8] for *Vigna mungo*, modified borax decahydrate extraction buffer [9] for lentil, modified Guanidine Thiocyanate (GITC) Lysis Buffer [10] for *Cajanuscajan*, Fruit-mate™ with Buffer RLT extraction method [11] for *Acacia koa* and *Leucaenaleucocephala* (tree legume). For non-legume plants, rapid CTAB based protocol [12] for grapevine, Trizol and Tris-HCl with β-mercaptoethanol protocol [13] for rice, Guanidine hydrochloride with sarcosyl based buffer for wheat have been used. All the above-mentioned protocols are crop as well as tissue specific and they do not work even on the related species [14]. Some commercial kits are also available such as Silica column based commercial plant RNA extraction kit (RNeasy ® Plant Mini Kit, Qiagen, Germany), Sigma Spectrum Plant Total RNA kit for RNA isolation in pigeonpea. These protocols are time consuming and cost incompetent.

Therefore, it becomes important to develop simple, rapid, efficient and cost-effective protocol that gave sufficient amounts of high-quality RNA from the different tissues of pigeonpea plant. So here we report conventionally used CTAB and Trizol based protocols with modifications for the isolation of total RNA.

## Materials and methods

### The modified CTAB method

**Material used.** Different tissues (leaf, flower and developing pod) of *Cajanuscajan* (cv. ICPL 87119) were obtained from the plants grown in the net house of ICAR-NIPB at Indian Agricultural Research Institute, New Delhi, India. The samples were immediately frozen in liquid nitrogen and stored at -80°C until the RNA extraction was done.

**Solutions and reagents.** CTAB-hexadecyltrimethylammonium bromide extraction buffer (2% CTAB, 2% PVP (polyvinylpyrrolidone), 100mM Tris-HCl (TRIS hydrochloride), 25mM EDTA (Ethylenediaminetetraacetic acid), lu NaCl and 2% BME, (β-mercaptoethanol), Phenol: Chloroform: Isoamyl alcohol (25:24:1), chilled Isopropanol, ice-cold 70% Ethanol and autoclaved 0.1% DEPC water.

**Protocol-1.** Different samples- leaf, flower and developing pods (100mg each) were ground to a fine powder in liquid nitrogen along with PVP using chilled mortar and pestle. The powder was transferred to 2ml micro centrifuge tubes containing 1ml pre warmed CTAB extraction buffer. The samples were mixed properly in the buffer by inverting the tubes and incubated in water bath at 65°C for 25 minutes. The samples were mixed after every 5 minutes. After completion of incubation, equal volume of Phenol: Chloroform: Isoamyl alcohol (25:24:1) was added and mixed vigorously by inverting the tubes for 6–8 times. The samples were centrifuged at 8000 rpm for 20 minutes at 4°C. The upper aqueous phase was transferred to a new 2ml tube and the Phenol: Chloroform: Isoamyl alcohol (25:24:1) step was repeated two more times. 800μl of chilled isopropanol was added to the transferred aqueous phase and mixed by inverting the tubes 6–8 times. The tubes were kept at -20°C for 1 hour for the RNA precipitation and centrifuged at 8,500 rpm for 30 minutes at 4°C. The supernatant was discarded and 1ml of ice-cold 70% ethanol was added to the pellet for washing. The tubes were centrifuged at 10,000 rpm for 5 minutes at 4°C. The washing step was repeated twice. The pellet was air dried for 10 minutes and dissolved in 20μl of autoclaved DEPC treated water.

## The modified Trizol method

**Material used.** Same materials were taken as used in above protocol-1

**Solutions and reagents.** Trizol (Tri-reagent) (Ambion, Life Technologies, California, USA, stored at 4˚C), chloroform, chilled isopropanol, 1.2M NaCl, 0.8M Sodium citrate, ice-cold 70% ethanol and autoclaved 0.1% DEPC (Diethyl pyrocarbonate) treated water were used.

**Protocol-2.** Different samples- leaf, flower and developing pods (100mg each) were ground to a fine powder in liquid nitrogen using chilled mortar and pestle. 2ml/100mg of Trizol were added in the mortar itself and left at room temperature (RT) till the sample completely thawed. The samples were transferred to 2ml micro-centrifuge tubes and centrifuged at 10,000 rpm for 10 minutes at 4˚C. The supernatant was transferred to new 2ml tubes. Then 400µl of chloroform was added and mixed for 60 seconds by inverting the tubes 6–8 times. The tubes were kept at 37˚C for 5 minutes. After the incubation the tubes were centrifuged at 10,000 rpm for 10 minutes at 4˚C. The supernatant was extracted with chloroform three more times. For the total RNA precipitation, 400 µl of isopropanol, 200 µl of 0.8M sodium citrate and 200 µl of 1.2M sodium chloride was added, mixed gently and kept at 37˚C for 5 minutes. The samples were incubated at -20˚C for 1 hour and centrifuged at 8,000 rpm for 30 minutes at 4˚C. The pellet was washed with 1ml of ice-cold 70% ethanol and centrifuged at 10,000 rpm at 4˚C for 5 minutes. The washing step was repeated twice. The pellet was air dried for 15 minutes and dissolved in 20 µl of autoclaved DEPC treated water.

**Analysis of RNA by Nanodrop spectrophotometer and agarose gel electrophoresis.** The RNA purity and concentrations were measured using NanoDrop-1000 vs 5.2 Spectrophotometer. The absorbance ratio, A260/A280 used to determine the protein contamination and A260/230 used to measure the polysaccharides contamination in RNA extractions. The RNA isolated from both the protocols was run on 1.5% denaturing formaldehyde agarose gel (100 ml). 1.5g of agarose was melted in 84 ml of DEPC treated autoclaved water. After the temperature dropped to around 60˚C, 10 ml of 10X MOPS (3-N-morpholino propane sulfonic acid) and 6 ml of 37% formaldehyde were added, mixed thoroughly and poured onto the horizontal gel casting tray. ~5µg (5µl) of RNA samples were mixed with equal volume of Thermo Scientific 2X RNA Loading Dye and denatured at 65˚C for 10 min followed by incubation for 5 minutes on ice. The samples were loaded carefully into the wells. The electrophoresis was carried out at 60V till the tracking dye reached the bottom of the gel. After electrophoresis gel was stained by overlaying it for 10 minutes with 1 µg/ml ethidium bromide to cover the gel. Finally, the RNA bands were visualized using gel documentation unit.

**DNase I treatment.** Prior to downstream application, 1µl of DNase I (2U/µl) (Thermo scientific) enzyme was added to 1µg of total RNA and incubated at 37˚C for 30 min to remove genomic DNA contamination. After incubation the enzyme was inactivated by incubating at 65˚C for 10 min in the presence of EDTA.

**Downstream application; cDNA synthesis and RT PCR.** For cDNA synthesis, 1µg of total RNA isolated from different tissues (leaf, flower and developing pod) was reverse transcribed into cDNA using Prime Script™ 1st strand cDNA Synthesis Kit (Takara) following manufacture's protocol. PCR amplification of cDNA synthesized was performed using specific primers (Forward-5′ GGAAGTAGACGAATTCAGGA3′ and Reverse-5′AGCACCTGCCCGATGAAG3′) for Serpin protease inhibitor gene with the amplicon length of 342bp. The PCR reaction was carried out in a total volume of 20µl reaction mixture containing, 2µl of 10X Taq buffer, 0.25µl of 1.25U Go Taq DNA polymerase (5U/µl), 1µl of 10mM dNTPs, 0.5µl of 0.5µM each primer, 2.5µl of 25mM MgCl2 and 1µl of cDNA as a template and in separate PCR reaction 1 µl of total RNA as template instead of cDNA to check the genomic

DNA contamination. PCR amplification cycle was optimized for Serpin protease inhibitor gene with following temperature profile:—Initial denaturation at 94˚C for 3min, 40 cycles (Denaturation at 94˚C for 30sec, annealing at 58˚C for 45sec, extension at 72˚C for 1min) followed by final extension at 72˚C for 6min. PCR amplified products were analysed by gel electrophoresis on 1.5% agarose gel in a TAE buffer and bands were visualized under gel documentation system.

## Results and discussion

Initially we tried to apply the widely used protocols like 1) CTAB [14], 2) Trizol as per the manufacturer's instructions and 3) CTAB-LiCl for RNA isolation from the pigeonpea plant. However, the quality and quantity of total RNA was not good and sufficient for the subsequent experiments as shown in the (Fig 1A–1C). Also, we failed to isolate sufficient quantity of RNA using commercially available RNA extraction kit. Although, these methods have been used successfully for the isolation of RNA from tissues of a wide range of plant species [12, 15–17], some species have proven difficult to isolate high-quality RNA with satisfactory yields. Additionally, these procedures increase the likelihood of co-purifying contaminants, which interfere in the subsequent applications [18, 19]. A prior study indicated that the modified CTAB procedure was effective for isolating RNA from various oil palm tissues [7]. To achieve the desired outcomes in the legume crop, pigeonpea, in the present study the most popular RNA extraction methods (CTAB and Trizol) were altered at different steps and evaluated by the quantity, purity, and integrity of the recovered RNA.

The CTAB extraction buffer protocol was modified, that we did not use 0.5g/l of spermidine in the extraction buffer, re-suspension buffer and also 10M LiCl. For the phase separation, the Phenol: Chloroform: Isoamyl alcohol (25:24:1) steps were performed at 8000 rpm for 20minat 4˚C instead of 10,000 rpm two times to make the supernatant very clear. RNA was precipitated by using 0.8ml of isopropanol and incubated at -20˚C for 1 hour instead of using ¼ volume of 10M LiCl and overnight incubation. After the precipitation the samples were centrifuged at 8,500 rpm for 30 minutes. The protocol [14] followed a single washing step with 70% cold ethanol, but in the reported protocol washing step was repeated two times to remove the excess of salts. The reported protocol required less time due to 1 hour isopropanol precipitation instead of overnight LiCl precipitation and it did not affect the quality of RNA.

The modified trizol protocol included addition of 2ml trizol/100mg of sample instead of 1ml trizol/100 mg for the complete RNA extraction. Trizol helps in maintaining RNA integrity

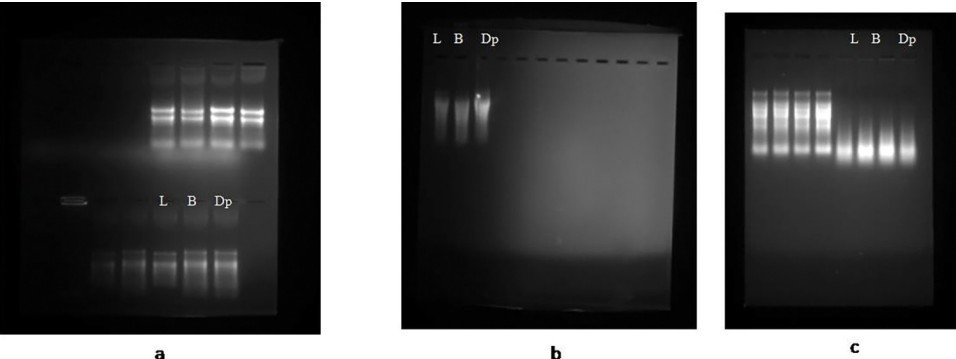

**Fig 1. Denaturing formaldehyde agarose gel electrophoresis of RNA isolated from three different tissues (L-Leaf, B-Bud, Dp-Developing pod) of pigeonpea using different RNA isolation methods (unmodified).** Fig 1A-CTAB method, Fig 1B-Trizol method, Fig 1C-CTAB-LiCl method.

due to highly effective inhibition of RNase activity during tissue homogenization while at the same time disrupting and breaking down cells and cell components. Due to the presence of excessive polyphenolic compounds and polysaccharide in pigeonpea, 2x (double) volume of trizol was added for more desirable disruption of cell components. In phase separation, chloroform step was carried out thrice to get rid of the proteins, trizol, lipids and DNA contamination. Different ranges of rpm (revolution per minute) were tried in the centrifugation step (10,000, 12,500 and15,000 rpm), but it was observed that a speed of 10,000 rpm for 10min was appropriate for extracting RNA from the pigeonpea samples. The precipitation step was carried out by adding 0.8 ml of isopropanol and the centrifugation step was carried out at a lower rpm i.e 8000 rpm for a longer time of 15–20 minutes which was most effective. Washing step was carried out twice by using 1ml of 70% cold ethanol in each step to remove the excess salts. The trizol (Ambion) based single step RNA isolation method was previously used for the RNA isolation from the pigeonpea plant and observed that trizol reagent played a better role in maintaining the RNA integrity but it was unable to precipitate large size rRNA 28S and 18S, may be because of less quantity of guanidinium isothiocyanate in the tri reagent [2].

The modified CTAB and trizol protocols yield sufficient and good quality of total RNA from all the tissues used in the study (leaf, flower and developing pod) ranged from 289to 422µg per gram of tissue. In contrast, unmodified protocols did not produce adequate amount of RNA, as the CTAB method produced RNA was partially degraded and 28S and 18s rRNA bands were not distinctly separated (Fig 1A). For the precise result of real time PCR, partially degraded RNA did not produce a reliable gene expression [17]. The RNA isolated by usual trizol method did not show any bands of 18S and 28S rRNA in the gel picture (Fig 1B), indicating that they were unable to precipitate high molecular weight RNA in the presence of high phenolic and carbohydrates compounds. The quality and purity of the isolated RNA was assessed byusingNanoDrop-1000 vs 5.2 Spectrophotometer by measuring absorbance at 260nm (measures for nucleic acid concentration), 280nm (measures for proteins) and 230nm (measure other contaminants such as polysaccharides and polyphenols, trizol, chaotropic salts) Fig 2.

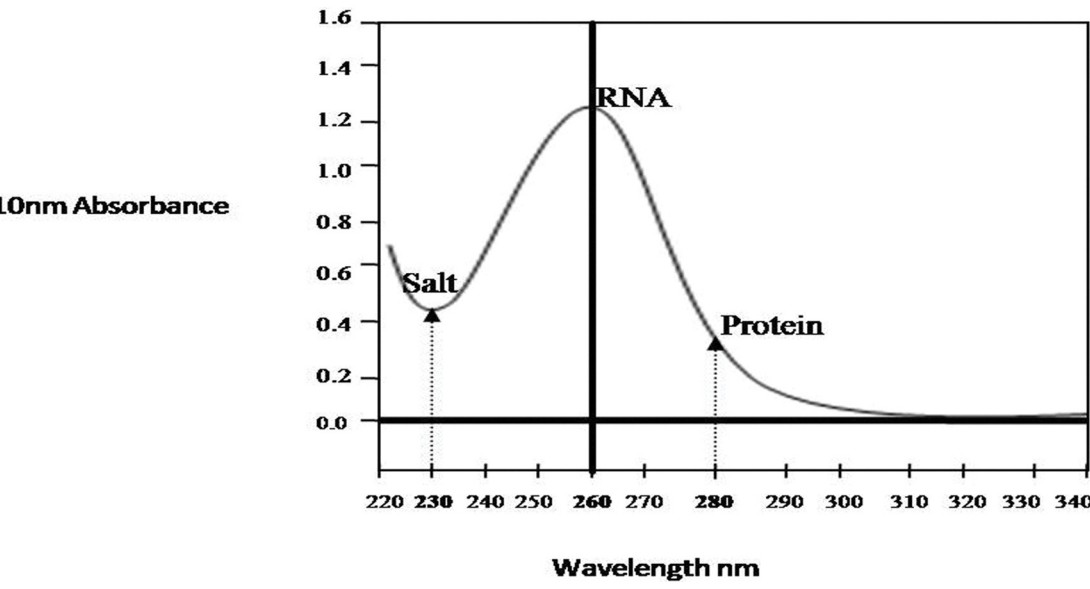

**Fig 2. The graph depicting the RNA purity measured by NanoDrop-1000 vs. 5.2 Spectrophotometer.** 230nm (measures for contaminants such as, trizol, chaotropic salts, carbohydrates and phenols), 260nm (measures for nucleic acid concentration), 280nm (measures for proteins).

**Table 1. Quantification of total RNA isolated from three different tissues of pigeonpea using modified CTAB and Trizol method using NanoDrop-1000 vs 5.2 Spectrophotometer.**

| Modified Methods | Plant samples | RNA Purity ratios | | RNA yield (ug per gram of tissue) |
|---|---|---|---|---|
| | | 260/280 | 260/230 | |
| CTAB | Leaf | 1.93±0.040 | 2.18±0.027 | 350±2.08 |
| | Bud | 1.91±0.076 | 2.13±0.145 | 329±2.64 |
| | Developing pod | 1.86±0.044 | 2.23±0.075 | 289±1.73 |
| Trizol | Leaf | 1.91±0.040 | 2.22±0.057 | 422±1.52 |
| | Bud | 1.98±0.076 | 2.11±0.049 | 374±0.95 |
| | Developing pod | 1.83±0.044 | 2.06±0.072 | 321±3.21 |

The purity of RNA ranged from 1.83 to 1.98 for the absorbance ratio A260/A280 indicating that the RNA was free of protein contamination. The A260/A230 ratio ranged from 2.03 to 2.23 showing that the RNA was devoid of contaminants (Table 1). The absorbance ratios 260/280more than 1.8 are accepted indicators of good quality RNA [20]. Distinct visible bands of high molecular weight rRNA 28S and 18S rRNA and low molecular weight rRNA was distinctly separated and observed without degradation or smear, indicating that the RNA was intact and free from the contaminants (Fig 3A modified CTAB and Fig 3B modified Trizol).

To utilize RNA in the downstream application, total RNA isolated from three different tissues (leaf, flower and developing pod) were treated with DNase Iand1ug of RNA were reverse transcribed to make cDNA. The standard PCR was performed with the Serpin protease inhibitor (342bp)gene specific primer and an amplicon of the desired length was successfully amplified. The RT-PCR results indicated that the RNA isolated using both the modified protocols were quantitatively sufficient and compliant for the downstream application (Fig 4, Lane 1–6). Hence the improved methods for the RNA isolation is rapid, efficient, easy and cost effective and can be used in the genetic studies in the molecular biology methods.

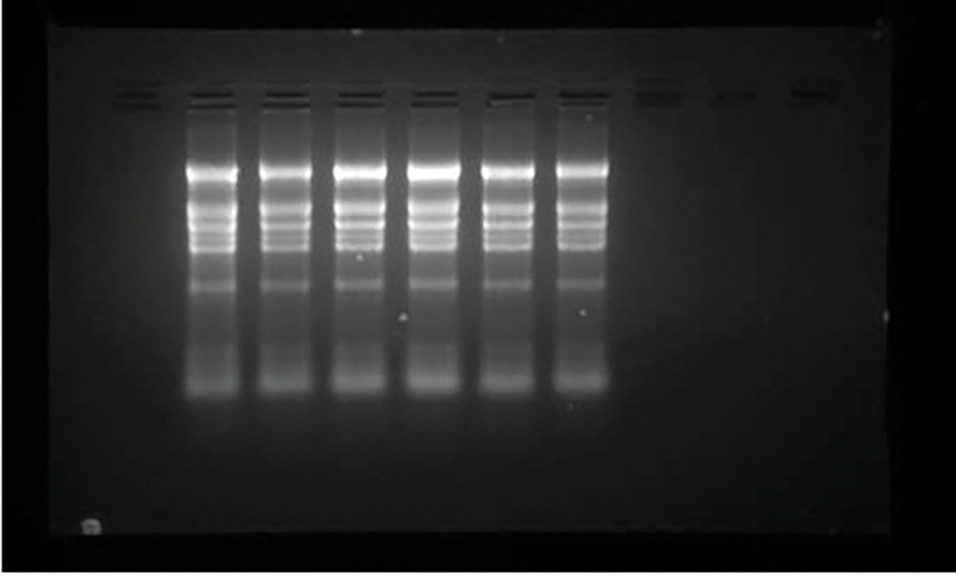

**Fig 3.** Denaturing formaldehyde agarose gel electrophoresis of RNA isolated from three different tissues (L-Leaf, B-Bud, Dp-Developing pod) of pigeonpea using modified CTAB (3a) and Trizol method (3b).

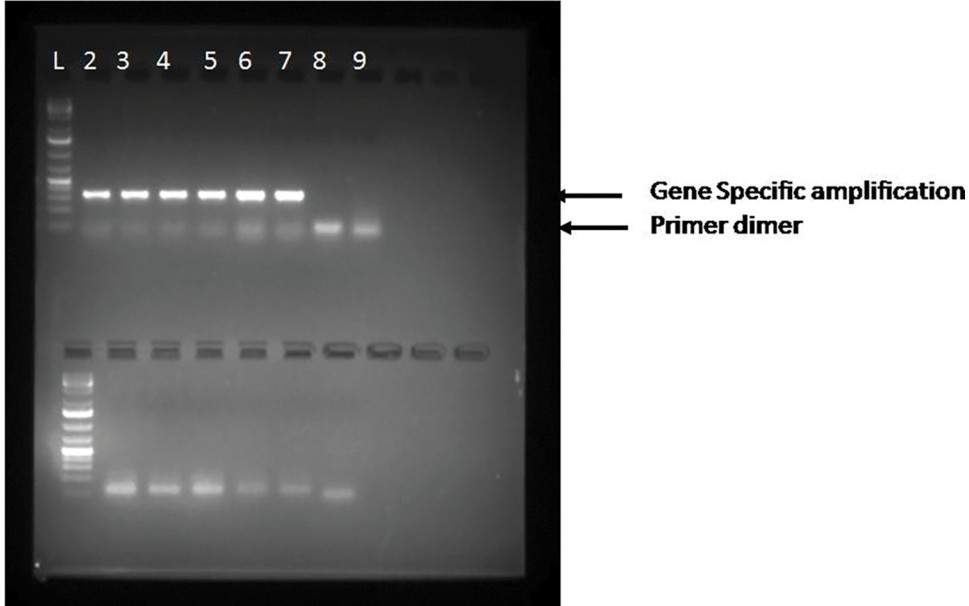

**Fig 4.** PCR amplification of Serpin protease inhibitor gene using cDNA prepared from RNA isolated by modified CTAB and Trizol method (Lane 1-1kb plus DNA Ladder, Lane 2–7 PI gene amplification, Lane 8-PCR amplification using RNA as the template, Lane 9-control with water.

## Conclusion

In the current study, two modified RNA extraction methods were explored and revealed to be effective in isolating high quality and quantity of RNA from varied tissues of *C. cajan*. Both of the methods are doable, affordable, and suitable for plants with high polyphenolic and polysaccharide content.

## Supporting information

**S1 Raw images.**
(PDF)

## Author Contributions

**Conceptualization:** Rekha Kansal.

**Data curation:** Pawan Mainkar, Deepanshu Jayaswal, Deepesh Kumar, Kuldip Jayaswall.

**Formal analysis:** Pawan Mainkar, Deepanshu Jayaswal, Deepesh Kumar, Kuldip Jayaswall, Sandeep Jaiswal, Sanjay Kumar.

**Investigation:** Deepanshu Jayaswal, Kuldip Jayaswall, Sanjay Kumar.

**Methodology:** Pawan Mainkar, Deepesh Kumar, Arvind Nath Singh.

**Resources:** Arvind Nath Singh, Rekha Kansal.

**Supervision:** Rekha Kansal.

**Validation:** Pawan Mainkar, Kuldip Jayaswall, Sandeep Jaiswal, Sanjay Kumar.

**Visualization:** Pawan Mainkar.

**Writing – original draft:** Pawan Mainkar, Deepanshu Jayaswal.

**Writing – review & editing:** Arvind Nath Singh, Sanjay Kumar, Rekha Kansal.

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
