## [Decision Letter · Decision Letter 0]

22 May 2023

PONE-D-23-05551Development of Modified CTAB and Trizol protocols to isolate High Molecular Weight (HMW) RNA from polyphenol and polysaccharides rich Pigeonpea (Cajanus cajan (L.) MillspPLOS ONE

Dear Dr. Kansal,

Thank you for submitting your manuscript to PLOS ONE. After careful consideration, we feel that it has merit but does not fully meet PLOS ONE’s publication criteria as it currently stands. Therefore, we invite you to submit a revised version of the manuscript that addresses the points raised during the review process.

We look forward to receiving your revised manuscript.

Kind regards,

Ramachandran Srinivasan, Ph.D.

Academic Editor

PLOS ONE

“The authors would like to thank the Department of Biotechnology, Government of Indiaand ICAR-NPTC project for providing the financial assistance”

Reviewers' comments:

Reviewer's Responses to Questions

**Comments to the Author**

1. Is the manuscript technically sound, and do the data support the conclusions?

Reviewer #1: No

Reviewer #2: Yes

2. Has the statistical analysis been performed appropriately and rigorously? 

Reviewer #1: N/A

Reviewer #2: Yes

3. Have the authors made all data underlying the findings in their manuscript fully available?

Reviewer #1: Yes

Reviewer #2: Yes

4. Is the manuscript presented in an intelligible fashion and written in standard English?

Reviewer #1: Yes

Reviewer #2: Yes

5. Review Comments to the Author

Reviewer #1: The article entitled "Development of Modified CTAB and Trizol protocols to isolate High Molecular Weight (HMW) RNA from polyphenol and polysaccharides rich Pigeonpea (Cajanus cajan (L.)" is well prepared. But the article doesn't say how the polyphenol and polysaccharides are removed from the sample. What are the chemicals/steps involved in the process of removal of polyphenol and polysaccharides.

Reviewer #2: The manuscript is good and appropriate to the journal, however the following comments need to be rectified before it is getting accepted for publication. The comments are given below:

1) Highlight the major outcome that differentiates the current protocol with the earlier reports.

2) How efficient the current protocol with CTAB- LiCl method as it provide higher quality and quantity?

3) More literatures are available for CTAB-Trizol for Plant RNA isolation. What is the need of your work?

2) Does this protocol is applicable to aquatic plants?

3) What are the advantages and limitations of already available CTAB and Tizol method from your view with current work?

5) What are the other protocols that are available for terrestrial plants RNA isolation ?

6) Why the author haven’t used RNA/DNA ladder in the FIgue 1&3 ?

7) Did you tried to isolate RNA from root sample using this protocol?

6. PLOS authors have the option to publish the peer review history of their article (what does this mean?). If published, this will include your full peer review and any attached files.

Reviewer #1: **Yes: **Gothandam Kodiveri Muthukaliannan

Reviewer #2: No

---

## [Author Response · Author response to Decision Letter 0]

19 Aug 2023

Response to editor:

We did not receive any specific fund for this work.

We have all the dataset in the MS only. No specific dataset is required out of the MS.

The attached image of the gel electrophoresis is cropped only to remove other parts outside of the gel. The well and primer dimer can be clearly seen in the image. However, if raw figures will be required, we will upload that also.

Response to reviewer 1 comments

Comment 1: The article entitled "Development of Modified CTAB and Trizol protocols to isolate High Molecular Weight (HMW) RNA from polyphenol and polysaccharides rich Pigeonpea (Cajanus cajan (L.)" is well prepared. But the article doesn't say how the polyphenol and polysaccharides are removed from the sample. What are the chemicals/steps involved in the process of removal of polyphenol and polysaccharides.

Response: Thanks for reviewing the manuscript critically that will indeed improve the quality of the MS. As per as the chemicals/steps are concerned, We have used (1) PVP with the liquid nitrogen during crushing of the samples (2) The 2X volume of Trizol was used (containing 2 times guanidium thiocyanate) (3) Phenol Chloroform Isoamylalcohol was used rather than only Isoamylalcohol (4) the precipitation was done for 1 hour rather than 24 h (5) Washing step was repeated to remove excess salts

Response to reviewer 2 comments

Comment 1: Highlight the major outcome that differentiates the current protocol with the earlier reports. 

Response: Thank you for the raising the outcome issues that will make the MS different to others. Following outcomes are in the MS.

(1) Yield sufficient and good quality of total RNA from all the tissues used in the study (leaf, flower and developing pod) ranged from 289 to 422μg per gram of tissue (Line 206, 207)

(2) Following the earlier protocols, RNA obtained were partially degraded and did not produce a reliable gene expression

(3) The RNA isolated by usual trizol method did not show any bands of 18S and 28S rRNA in the gel picture (Fig 1b), indicating that they were unable to precipitate high molecular weight RNA in the presence of high phenolic and carbohydrates compounds (Line 211 to 214).

In the concerned paragraph, all outcomes are mentioned. (Line number 206-224)

Comment 2: How efficient the current protocol with CTAB- LiCl method as it provides higher quality and quantity?

Response: The current method is more productive than existing CTAB-LiCL method, as it mentioned in terms of higher quality and quantity of RNA particularly in case of pigeonpea. However, the existing CTAB-LiCL method could also be useful for other crops.

Since a 1-hour isopropanol precipitation was utilised instead of an overnight LiCl precipitation, the proposed method required less time compared to CTAB-LiCl method while maintaining the same level of RNA quality and quantity. The reported protocols washing step was performed twice to remove excess salts, as opposed to the CTAB-LiCl method single washing step using 70% cold ethanol.

Comment 3: More literatures are available for CTAB-Trizol for Plant RNA isolation. What is the need of your work?

Response: I respectfully agree with the concerned raised. As per mentioned statements and figures in the manuscript, the existing methods were used to isolate RNA from Pigeonpea but somehow, the satisfactory results were not obtained. Therefore, the study was conducted to develop this modified method of RNA isolation.

Comment 4: Does this protocol is applicable to aquatic plants?

Response: We have not tried to isolate RNA from aquatic plants following this protocol as our research was on Pigeonpea but in our opinion, it could be used for aquatic plants.

Comment 5: What are the advantages and limitations of already available CTAB and Tizol method from your view with current work?

Response: Thanks. The advantage of already available CTAB and Tizol method will be as per the protocol tested on the concerned individuals. Here, in our case, the existing protocol was not too favorable, so this is the limitation of the existing protocol. Additionally, the modified steps and chemicals that we have used were limiting in the already available CTAB and Tizol method that made our study different from them. 

The modified CTAB and trizol protocols yield sufficient and good quality of total RNA from all the tissues used in the study (leaf, flower and developing pod) ranged from 289 to 422 μg per gram of tissue. In contrast, unmodified protocols did not produce adequate amount of RNA, as the CTAB method produced RNA was partially degraded and 28S and 18S rRNA bands were not distinctly separated. The RNA isolated by usual trizol method did not show any bands of 18S and 28S rRNA in the gel picture, indicating that they were unable to precipitate high molecular weight RNA in the presence of high phenolic and carbohydrates compounds. The improved methods for the RNA isolation are rapid, efficient, easy and cost effective and can be used in the genetic studies in the molecular biology methods. Additionally, the modified protocols removed the excess of salt and co-purifying contaminants, which interfere in the subsequent applications and these protocols requires less time compared to conventional method.

Comment 6: What are the other protocols that are available for terrestrial plants RNA isolation?

Response: SDS-phenol based, CTAB method, Trizol method and CTAB-LiCl methods are available for terrestrial plants RNA isolation.

Comment 7: Why the author haven’t used RNA/DNA ladder in the FIgue 1&3?

Response: Figures 1 and 3 in the manuscript both showed RNA. The most important factors in determining the quality and amount of the material during RNA gel electrophoresis are the size of the bands (28S and 18S), the intensity of the bands, and the presence/absence of smears. Consequently, RNA/DNA ladder has not been taken into account for both figures.

Comment 8: Did you tried to isolate RNA from root sample using this protocol?

Response: We are sorry to say that we have not tried to isolate RNA from root sample using this protocol but in our opinion it will work for sure as we are using PVP during crushing of the sample with liquid nitrogen so it will while using hard tissues like roots.

---

## [Decision Letter · Decision Letter 1]

10 Sep 2023

Development of Modified CTAB and Trizol protocols to isolate High Molecular Weight (HMW) RNA from polyphenol and polysaccharides rich Pigeonpea (Cajanus cajan (L.) Millsp

PONE-D-23-05551R1

Dear Dr. Rekha Kansal,

We’re pleased to inform you that your manuscript has been judged scientifically suitable for publication and will be formally accepted for publication once it meets all outstanding technical requirements.

Kind regards,

Ramachandran Srinivasan, Ph.D.

Academic Editor

PLOS ONE

Additional Editor Comments (optional):

Reviewers' comments:

Reviewer's Responses to Questions

**Comments to the Author**

1. If the authors have adequately addressed your comments raised in a previous round of review and you feel that this manuscript is now acceptable for publication, you may indicate that here to bypass the “Comments to the Author” section, enter your conflict of interest statement in the “Confidential to Editor” section, and submit your "Accept" recommendation.

Reviewer #1: All comments have been addressed

Reviewer #2: All comments have been addressed

2. Is the manuscript technically sound, and do the data support the conclusions?

Reviewer #1: Yes

Reviewer #2: Yes

3. Has the statistical analysis been performed appropriately and rigorously? 

Reviewer #1: N/A

Reviewer #2: Yes

4. Have the authors made all data underlying the findings in their manuscript fully available?

Reviewer #1: Yes

Reviewer #2: Yes

5. Is the manuscript presented in an intelligible fashion and written in standard English?

Reviewer #1: Yes

Reviewer #2: Yes

6. Review Comments to the Author

Reviewer #1: The revised article entitled "Development of Modified CTAB and Trizol protocols to isolate High Molecular Weight (HMW) RNA from polyphenol and polysaccharides rich Pigeonpea (Cajanus cajan (L.)" has been addressed all the comments, and it can be accepted for publication.

Reviewer #2: The author have addressed all the queries which was raised earlier and the transformation of manuscript after the review comments is more appropriate for publication.

7. PLOS authors have the option to publish the peer review history of their article (what does this mean?). If published, this will include your full peer review and any attached files.

Reviewer #1: No

Reviewer #2: No

---

## [Editor Report · Acceptance letter]

30 Nov 2023

PONE-D-23-05551R1 

Development of Modified CTAB and Trizol protocols to isolate High Molecular Weight (HMW) RNA from polyphenol and polysaccharides rich pigeonpea (*Cajanuscajan* (L.) Millsp. 

Dear Dr. Kansal:

I'm pleased to inform you that your manuscript has been deemed suitable for publication in PLOS ONE. Congratulations! Your manuscript is now with our production department. 

Kind regards, 

on behalf of

Dr. Ramachandran Srinivasan 

Academic Editor

PLOS ONE